# SOI: Scaling down computational complexity by estimating partial states of the model

## Abstract

Consumer electronics used to follow the miniaturization trend described by Moore's Law. Despite the continuous growth in the processing power of Microcontroller Units (MCUs), MCUs used in the smallest appliances are still not capable of running even moderately big, state-of-the-art artificial neural networks (ANNs). Deploying ANNs on this class of devices becomes even more challenging when they are required to operate in a time-sensitive manner, as the model's inference cannot be distributed over time. In this work, we present a novel method called Scattered Online Inference (SOI) that aims to reduce the computational complexity of ANNs. SOI is developed based on the premise that time-series data is continuous and/or seasonal, and so are the model's predictions. This enables extrapolation leading to processing speed improvements, especially in the deeper layers of the model. The application of strides forces the ANN to produce more general inner partial states of the model, as they are based on a higher number of input samples which lie further apart from each other. As a result, SOI allows for skipping full model recalculation at each inference by performing the strictly necessary operations only. We present two possible patterns of inference achievable with SOI - Partially Predictive (PP) and Fully Predictive (FP). For the audio separation task, we achieved a 64.4% reduction in computational complexity at the cost of 9.8% of SI-SNRi for the PP variant, and a 41.9% reduction at the cost of 7.70% SI-SNRi with the FP variant. Moreover, the FP variant reduces inference time by additional 28.7%. Similar results are also presented for the acoustic scene classification task with a model based on the GhostNet architecture.

## 1 Introduction

Moore's Law (Moore, 1998) predicts that the number of transistors on a microchip will double every 2 years. As the number of transistors increases, the computational capacity of systems-on-chip (SoCs) also grows. Accompanied by the further miniaturization of components, the increased complexity of SoCs allows for the production of more compact appliances with equal or higher capabilities.

Despite the continuous development of more advanced hardware, the smallest appliances still cannot fully benefit from increasingly popular neural-based solutions, as they are not able to run them efficiently. Examples of such appliances include wireless in-ear headphones, smart watches, and AR glasses. Furthermore, the size of state-of-the-art neural networks is growing at much faster rate than that described by Moore's Law (Xu et al., 2018). Adding to the challenge, according to researchers (Leiserson et al., 2020), Moore's Law is starting to decelerate due to the physical constraints of semiconductors, and the "room at the bottom" is depleting.

Currently available neural network technologies enable machines to outperform humans in numerous applications in terms of measured performance (Nguyen et al., 2020; Bakhtin et al., 2022; Schrittwieser et al., 2020). However, despite this achievement, the same models fall significantly short when it comes to energy efficiency compared to humans. The human brain consumes a mere 20 Watts of power (Laughlin et al., 1998; Sengupta & Stemmler, 2014; Balasubramanian, 2021) and according to Xu et al. (2018) it is estimated to be over five orders of magnitude more energy efficient than modern neural networks.

This disparity in energy efficiency can be attributed to the common pursuit of the highest model quality in the literature, showcasing the full capabilities of the developed technology, often at the expense of efficiency. This behavior is justifiable due to the ease of comparing different solutions using well-defined metrics that are independent of the hardware and software. Conversely, estimating a model's energy efficiency is a more complex task, influenced by various factors including the running environment.

However, this trend within our community may prove restrictive, particularly for applications like real-time systems. These applications naturally demand optimal performance alongside high-efficiency solutions, rendering most current Deep Neural Networks (DNNs) impractical for such tasks. Furthermore, due to the substantial discrepancy between assumptions for high-efficiency on-device solutions and high-performing monolithic models, compressing these models may not be a viable means of applying state-of-the-art DNNs to real-time tasks.

The importance of neural system efficiency is also increasingly significant from ecological and economic standpoints (Lacoste et al., 2019; Patterson et al., 2021).

## 1.1 RELATED WORKS

The Short-Term Memory Convolution (STMC) (Stefański et al., 2023) was devised to enhance the efficiency of Convolutional Neural Networks (CNNs) inference by reducing computation requirements at each step, eliminating the need to recompute prior states. The authors achieved a notable 5.68-fold reduction in inference time and a 30% decrease in memory consumption compared to the Incremental Inference method (Romaniuk et al., 2020). STMC enables the conversion of a standard CNN model, which typically requires an input of size at least as large as its receptive field, into a model capable of processing a single input frame at a time, akin to Long-Short Term Memory networks (LSTM). The SOI method is built upon the STMC foundation, offering distinct treatment of strided layers and yielding a compelling new *inference pattern*[1]. A brief overview of STMC can be found in appendix A.

Routing methods constitute a popular category of algorithms tailored to optimize the inference process of Recurrent Neural Networks (RNNs). In the field of Natural Language Processing (NLP), Yu et al. (2017) introduced an approach that involves traversing segments of the computational graph. This traversal is guided by decisions made by a reinforcement learning model following the processing of a fixed number of words. Another notable contribution by Campos et al. (2018) yielded an algorithm capable of selectively bypassing partial state updates within an RNN during inference, influenced by the input's length. In NLP terms, this concept can be compared to the mechanism of skipping words.

The research by Jernite et al. (2017) introduced a distinct method to regulate computation within a recurrent unit. This method relied on a scheduler unit that facilitated partial updates to the network's state, only when the computational budget limit was reached. Meanwhile, Seo et al. (2018) proposed an approach referred to as "word skimming". In this approach, the authors designed both small and large RNN models that could be interchangeably utilized for inference through the utilization of Gumbel softmax. The exploration of hybrid techniques combining jumping, skimming, and skipping was further advanced by Hansen et al. (2019), who published additional solutions in this direction.

The RNN routing methods have found successful applications in CNNs as well. Wang et al. (2018) introduced an approach that enables a model to learn a policy for skipping entire convolutional layers on a per-input basis. A similar concept, involving adaptive inference graphs conditioned on image inputs, was put forth by Veit & Belongie (2018). Additionally, several authors have contributed methods for early-exit during CNN inference (Bolukbasi et al., 2017; Teerapittayanon et al., 2016; Huang et al., 2017). In these methods, the network is trained to skip portions of the computational graph towards the final stages, based on the characteristics of the input.

Other commonly employed methods for model optimization include pruning (LeCun et al., 1989) and quantization (Gray & Neuhoff, 1998). Importantly, both of these methods are not mutually exclusive and can coexist alongside routing methods within a single neural network.

---

[1] By the term of an inference pattern we mean a full computational graph of a single inference or of repeating sequence of inferences if they influence the computational graphs of the following model executions as in case of SOI or STMC.

## 1.2 Novelty

In this study, we introduce a method for reducing the computational cost of a CNN model while incurring only a negligible decrease in the model's performance. Importantly, these reductions are achieved with minimal alterations to the architecture, making it suitable for various tasks where factors such as energy consumption or time are of paramount importance.

Our approach involves the conversion of a conventional CNN model, initially trained to process segments of time-series data with arbitrary lengths in an offline mode, into a model that processes the data element by element, enabling real-time usage. Notably, our method builds upon the STMC technique studied in prior research (Stefański et al., 2023). STMC is designed to perform each distinct operation exactly once. SOI extends this concept by omitting some operations related to strided convolutions in a structured manner.

Strides are commonly employed in CNNs for offline inference, where complete data is available, their application becomes challenging in online scenarios where data must be processed element by element. From the standpoint of online inference, the utilization of strided convolutions can be likened to introducing *gaps* in the output sequence. To tackle this concern in the context of Scattered Online Inference (SOI), our approach involves training models to predict the omitted values, effectively filling these gaps. Consequently, during the encoding phase, strided convolutions act as data compressors, while the decoder undertakes decompression via extrapolation. This entire scheme emulates an inference pattern reminiscent of RNNs, while capitalizing on the advantages of CNN models, including ease of training and controllability.

In this study, we introduce a novel method named Scattered Online Inference (SOI), which is based on the following key principles:

- The reduction of computational complexity is achieved through the implementation of partial predictions of the network's future state.

- SOI operates as a two-phase system. The initial phase involves compressing data within the time domain using strided convolutions. The subsequent phase focuses on data reconstruction, employing the most suitable extrapolation scheme.

- SOI enables a RNN-like inference pattern within CNNs by leveraging the capabilities of STMC layers.

- The method preserves the causal nature of the optimized network architecture.

- SOI's applicability is confined to a single-frame online inference, aligning with the limitations of the STMC method. Additionally, it necessitates the incorporation of skip connections to update the network's output following each inference.

## 2 Methods

When processing a time-series in online mode, the model goes through each incoming element[2] separately. In this paper we refer to such an event as *inference*.

An improvement in computation complexity is achieved by introducing the partially predictive strided convolutions adapted for online inference, as well as by avoiding the redundant computations done during previous inferences as in STMC study. Therefore, after each inference, the results which would be recalculated in subsequent runs are cached. We refer to such cacheable outputs as a *partial state* of the network.

It is easy to observe that using strides reduces layer's output size by a factor equal to the length of the stride. However, when performing online processing, our interest lies in the latest (rightmost) elements of the output and intermediate tensors. These are affected by the undesired edge effects. Typically, CNN networks are trained to accept input sizes corresponding to the arrangement of stride and pool operations. To cover online inference without repeating computations, a network should be capable of accepting input one element larger than the previous one. Our method systematically

---

[2]We define an "element" as a chunk of data of arbitrary size, including the smallest possible unit - a singular data point in a time series. The size of the incoming data chunk depends on the specific system being used.

addresses this requirement, allowing us to contain the "edge effect" and limit its influence on the output quality.

## 2.1 SCATTERED ONLINE INFERENCE

Scattered Online Inference (SOI) is a method which modifies the inference pattern of a network to skip the recalculation of certain layers in a predetermined pattern. This is achieved through the use of strided convolutional layers and extrapolation. Both operations are exclusively applied along the time axis. In this study, we employed a basic form of extrapolation, where the last known value is used to replace the missing ones[3]. To facilitate a better understanding of the SOI algorithm, in Figure 1, we define three types of convolutional layers utilized in our method. For comparison purposes, Figure 1 also includes standard convolution and strided convolution.

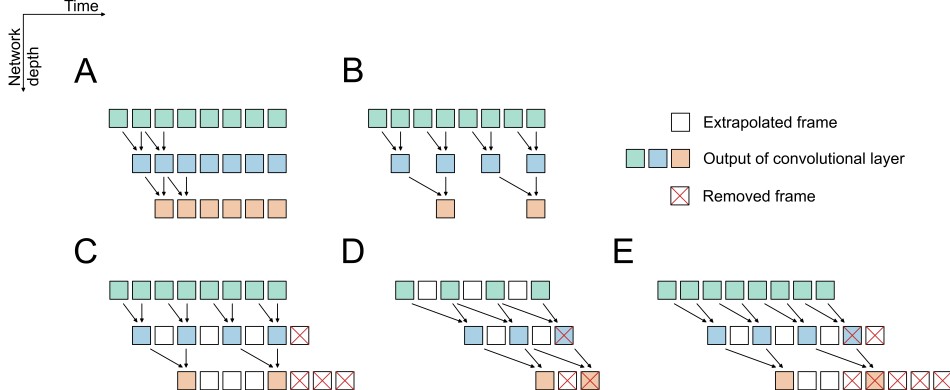

Figure 1: Convolutional operations. For visualization purposes we show data as frames in time domain. A) Standard convolution. B) Strided convolution. C) Strided-Cloned Convolution. D) Shifted convolution. E) Shifted Strided-Cloned Convolution.

Strided-Cloned Convolution (S-CC) performs a 2-step operation. Firstly, it applies a strided convolution to the input in order to compress the data. In the second step it fills the gaps created by striding using any form of extrapolation. In our experiments we extrapolate by simply copying a previous frame. The copied frame is then aligned with a future frame relatively to its input. In practice, we split the stride and extrapolation operations into different layers which results in optimization of computational complexity between those layers. Because of that we will refer to this as S-CC pair.

Shifted Convolution (SC) shifts data in time domain after the convolution thus creating additional predicted network states. This layer may be used for additional latency reduction.

Shifted Strided-Cloned Convolution (SS-CC) is a combination of S-CC pair and SC layer which is needed if we want to do both of them at the same point of the network. In our experiments we extrapolate output vector first and then apply data shifting to reduce size of introduced partial state prediction.

SOI can be divided into two types depending on how prediction is handled. These two types have significantly different inference patterns (Fig. 2).

**Partially Predictive (PP)**  In this type of SOI, we do not introduce any additional predictive components in the network. This implies that after compression, the most recent frame stores information for the current output frame and the future output frame. This type of SOI uses S-CC pairs only. This configuration results in only one inference (the first one), which updates all network states, while all subsequent ones use relevant buffered partial network states. Although hybrid setups involving more full passes are viable, they fall outside the scope of this paper. It's important to note that this mode does not reduce peak computational complexity, but rather the average computational complexity.

---

[3]More sophisticated methods were tested as well and are presented in supplementary materials.

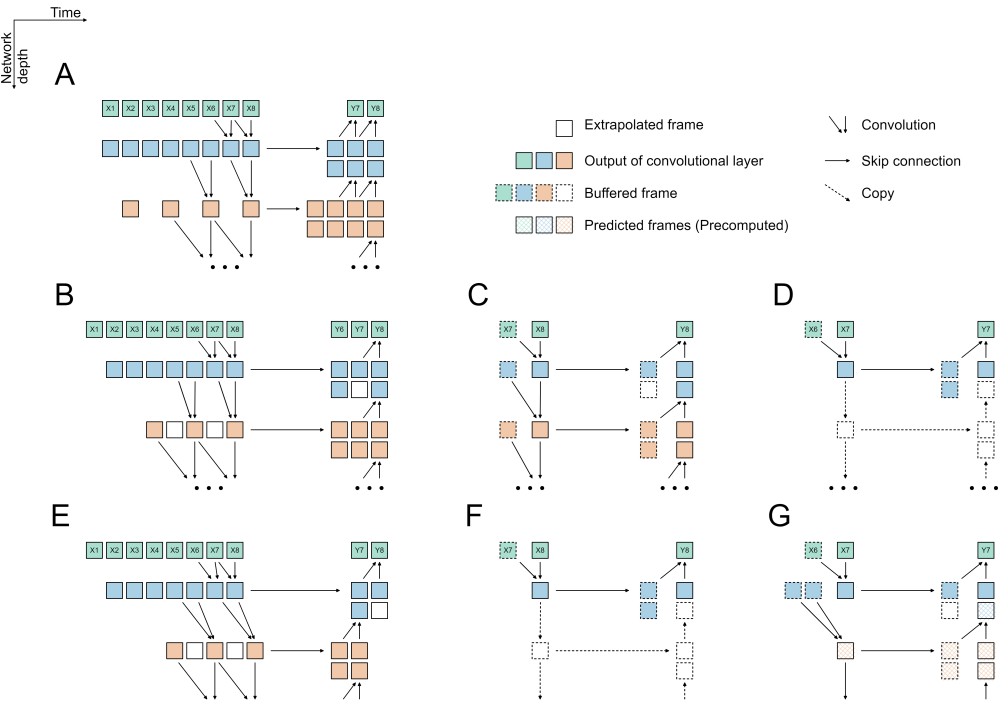

Figure 2: Inference patterns of each type of SOI based on U-Net architecture. A) Unmodified causal U-Net. B) Partially predictive (PP) SOI. C) Even inference of PP. D) Odd inference of PP. E) Fully predictive (FP) SOI. F) Even inference of FP. G) Odd inference of FP.

**Fully Predictive (FP)**   In this type of SOI, we introduce additional predictive components to the network. Compared to PP SOI, the most recent frame does not store any information about the current output frame, but rather about two future ones, hence the name "Fully Predictive", as only the future output is calculated. This is a more challenging task than PP SOI, but it can significantly decrease the latency of the model. This mode utilizes both S-CC pairs and SC layers. It optimizes both peak computational complexity and latency because it allows some inferences to be predicted in full. The fully predicted inference, in contrast to the regular inference which requires newly collected input, operates only on already processed data and can calculate relevant network states while the system awaits the new data, reducing the amount of computation required when the new data arrives.

Both of these types of SOI can be combined. This occurs when we first introduce PP SOI compression and then after some number of layers, we introduce an additional shift in the time axis after which the model can be treated as FP SOI.

## 2.2 MATHEMATICAL FORMULATION

Let us assume that an input of the model is represented by a 1D time series vector $X \in \mathbb{R}^N$ composed of $N$ samples. Additionally let's say that a network is composed of 5 convolutional layers and output vector $^l Y$ for $l$-th layer is of the same shape as the input. Each convolutional layer has a 1D kernel $h_l \in \mathbb{R}^{M_l}$ which can be represented by a Toeplitz matrix $H_l \in \mathbb{R}^{N \times (N-M_l+1)}$. We get:

$$^l Y = H_l \cdot {}^l X^T \tag{1}$$

After which we apply activation function $\sigma$ and get the input for the next layer:

$$^{l+1} X = \sigma(^l Y) \tag{2}$$

We use $^l X_t$ to denote a segment of $X$ ending in time $t$ of a length matching the context it is used in (e.g. assuming $h_l$ denotes a convolutional kernel of layer $l$ and provided with the context $h_l \cdot {}^l X_t$, $^l X_t$ has $M_l$ elements to match the kernel size).

By using the STMC inference pattern we perform inference for each element of $X$ separately and reuse past partial states of each layer to fully reconstruct the receptive field of the network which can be represented as follows:

$$^{l+1}X_t = \left(^{l+1}X_{t-1} \Big|_t \sigma(h_l \cdot {}^l X_t^T)\right) \tag{3}$$

where $(\cdot \mid_t \cdot)$ represents the concatenation of vectors in time axis.

If we use a convolutional layer with a stride of 2 as our second layer, then in the standard pattern, the size of the output vector of this layer is halved compared to its input. Consequently, every subsequent layer also has its input and output tensors reduced to half the size. We can restore the output to its original size by, for instance, applying transposed convolution. Let's assume that we apply transposed convolution in the 4th layer of our network. In comparison to our initial plain network, the new strided network will have the same number of operations in both the first and last layers. The 2nd, 3rd, and 4th layers will each have half the computations as before. In the 2nd layer, this reduction is due to the application of stride. In the third layer, it is a result of the smaller input size. Similarly, in the 4th layer, if we disregard multiplications by zero.

When employing the STMC inference method, it's anticipated that each layer should process a single element and produce a single output. If we apply a similar stride and transposed convolution pattern without any additional modification, we'll face difficulties in reconstructing the output. Strided convolution would provide output values for even-numbered inferences (assuming a stride size of 2). However, during odd inferences, the 4th layer (transposed convolution) would require an upcoming even-numbered inference value, which would not yet be available. The authors of STMC propose a solution where every inference is treated as even-numbered, maintaining separate states for odd and even input positions. However, this pattern presents a challenge due to the exponential increase in the number of states for each added strided convolution.

SOI addresses this issue by removing the necessity of storing additional states, albeit at a cost to measured performance. For instance, in our network, we achieve this by abstaining from calculating the outputs of the 2nd, 3rd, and 4th layers during every even inference. To maintain causality, the transposed convolution output must be temporally shifted to produce either current and future frames or solely future frames, depending on the chosen SOI inference mode. Additionally, we advocate for the use of a skip connection between the input of the strided convolution and the output of the transposed convolution to update deeper layers of the network with information about the current data. This operation aims to minimize the influence of data forecasting on the optimized part of the network.

To formally describe partially predictive SOI, let's assume that the network contains layer $l_d$ with a stride of size $2^4$ and layer $l_u$ which reverses the downsampling performed by $l_d$. Typically, $l_d$ would be in the encoder part, while $l_u$ would be in the decoder.

The downsampling layer only returns a new element for even-numbered inferences. It's important to note that until the upsampling layer is reached, there is no need to perform any further computations when $t$ is odd. Namely for $l$ such that $l_d < l \le l_u$:

$$^{l+1}X_t = \begin{cases} \left(^{l+1}X_{t-1} \Big|_t \sigma(h_l \cdot {}^l X_t^T)\right), & \text{if } t \text{ is even} \\ ^{l+1}X_{t-1}, & \text{if } t \text{ is odd} \end{cases} \tag{4}$$

At layer $l_u$ we reconstruct the output by duplicating the convolution output.

$$^{l_u+1}X'_t = \left(^{l_u+1}X'_{t-1} \Big|_t \sigma(h_{l_u} \cdot {}^{l_u} X_t^T)^T\right) \tag{5}$$

Note that $^{l_u}X_{2s}^T = {}^{l_u}X_{2s+1}^T$.

Current frame

Buffered frame

Copied frame

Removed frame

Figure 3: SOI PP inference pattern.

[4]The choice of 2 is for notational simplicity. The same derivation can be applied for an arbitrary stride size.

We then concatenate the output with data from the skip connection ($(\cdot \mid_c \cdot)$ represents the concatenation of vectors in channel axis).

$$^{l_u+1}X_t = \left(^{l_u+1}X'_t \mid_c \, ^{l_d}X_t\right) \tag{6}$$

In the example above we traded inference over $l_u - l_d$ layers at the cost of inference over additional channels from skip connection concatenation. The additional cost of skip connection does not take effect in architectures where skip connections naturally exist like U-Net which we use as one of the examples in this study.

For fully predictive variant we modify equation (5) and add a shift in time axis.

$$^{l_u+1}X'_t = \left(^{l_u+1}X'_{t-1} \mid_t \, \sigma(h_{l_u} \cdot \, ^{l_u}X^T_{t-1})^T\right) \tag{7}$$

SOI can be seen as a form of forecasting where instead of training the model to predict output for input yet unknown, we predict partial states of the network. This is because preserving causality in the strided convolution forces us to predict a convolution results when the next second input is not yet available.

Multiple SOI layers may be used within a single network.

## 3 EXPERIMENTS

### 3.1 SPEECH SEPARATION

#### 3.1.1 EXPERIMENTAL TASK

We selected speech separation as our first experimental task. The choice of the task is dictated by our current research interests and the potential benefits of fast online inference. In this task we focus on separating clean speech signal from noisy background. In literature this task is also referred to as speech denoising or noise suppression.

#### 3.1.2 BASE MODEL, TRAINING PROCESS AND DATASET

For this experiment we adopted the U-Net architecture as it is widely used for this specific task and inherently has skip connections which will allow for applying SOI inference pattern without substantial alterations. Our model is composed of 7 encoder and 7 decoder layers, each comprising STMC/tSTMC, batch norm and ELU activation layers. Each model was trained for 100 epochs using Adam optimizer with initial learning rate of 1e-3. We trained each model on a single Nvidia P40 GPU 5 times and reported the average SI-SNRi. The mean training time of a single model amounted to about 14 hours. The Deep Noise Suppression (DNS) Challenge - Interspeech 2020 dataset (Reddy et al., 2020), licensed under CC-BY 4.0, was used for both training and evaluation purposes. For training, we used 16384 10s samples without any form of augmentation and for both validation and test sets we used 64 samples with similar setup to the training set.

**Position of S-CC pair** By introducing S-CC pair to the network we are enforcing data predictiveness of the network. The exact number of the predicted future partial states of the model depends on the position of S-CC pair and number of those pairs within the network. In addition it is worth noting that larger amount of predicted partial states of the network leads to higher reduction of computational complexity. In this experiment we test every position of S-CC pairs while applying up to two such modules to the model's architecture.

**Position of SS-CC pair** SS-CC pair introduces additional shift in time axis compered to S-CC pair. In this experiment we alter the position of SS-CC pair within the network and also separately alter the position of S-CC pair and time shift which might be consider as a hybrid of partially and fully predictive pattern.

Additional results for this task can be found in Appendix B where we tested influence of strided convolution on predictive inference and Appendix C where we tested different extrapolation methods.

## 3.2 ACOUSTIC SCENE CLASSIFICATION

### 3.2.1 EXPERIMENTAL TASK

Acoustic scene classification (ASC) is our second task of choice. The goal of the task is to estimate the location where the specific sample was recorded. This task is commonly considered as an auxiliary problem in various online scenarios such as selection of bank of denoising filters.

### 3.2.2 MODELS AND TRAINING PROCESS

For all our tests in ASC task we used GhostNet architecture (Han et al., 2020). Our baseline model is the original architecture with "same" padding making it not applicable in online scenario. STMC model changes the padding to manually-applied padding in left-most (oldest) side of data and applies STMC inference pattern. SOI model adds upsampling after each processing block and skip connections between downsampling/upsampling layers.

Models were trained on a single Nvidia P40 GPU for 500 epochs using Adam optimizer with initial learning rate of 1e-3. We tested 7 different model sizes for all 3 variants - Baseline, STMC and SOI. Each test was repeated 5 times. We used the TAU Urban Acoustic Scene 2020 Mobile dataset (Heittola et al., 2020) for both training and validation.

## 4 RESULTS

### 4.1 SPEECH SEPARATION

**Partially predictive**   Results of partially predictive SOI in speech separation task are shown in figure 4. We tested variants with a single S-CC layer ("S-CC") and two S-CC layers ("2xS-CC $X$"). In the latter case, the experiments are grouped by the position of the first S-CC layer ($X$). Value of SI-SNRi metric is highly dependant on the position of S-CC layers. Generally, the earlier the S-CC layer is introduced the lower SI-SNRi but higher complexity reduction. This phenomena may be explained by the difficulty of the partial state prediction task. All layers subsequent to the S-CC layer are required to predict twice the number of elements compared to the layers without having the S-CC as an prior layer. Overall, with SOI we can achieve linear dependency between computation cost and SI-SNRi up to 64% of complexity reduction (with linearity factor of 0.001 dB SI-SNRi per 1 MMAC/s (0.017 dB SI-SNRi per 1% of reduction)). A significant SI-SNRi drop is observed if S-CC layer is introduced too early. In table 1 we present selected results from the whole experiment.

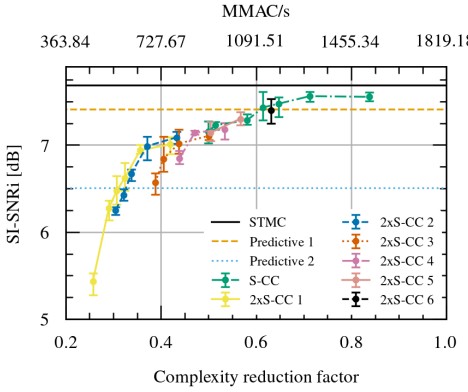

Figure 4: Results of speech separation experiment with PP SOI.

Table 1: Selected results from experiments with PP SOI for speech separation.

| Model | SI-SNRi (dB) | SI-SNRi retain (%) | Complexity retain (%) | Complexity (MMAC/s) |
|---|---|---|---|---|
| STMC | **7.69** $^{+0.06}_{-0.08}$ | **100.0** | 100.0 | 1819.2 |
| Predictive 1 | 7.41 $^{+0.07}_{-0.09}$ | 96.3 | 100.0 | 1819.2 |
| Predictive 2 | 6.51 $^{+0.03}_{-0.08}$ | 84.7 | 100.0 | 1819.2 |
| S-CC 2 | 7.23 $^{+0.04}_{-0.05}$ | 94.0 | 51.4 | 935.2 |
| S-CC 5 | 7.47 $^{+0.07}_{-0.15}$ | 97.2 | 64.8 | 1178.7 |
| S-CC 7 | 7.55 $^{+0.05}_{-0.05}$ | 98.2 | 83.8 | 1524.3 |
| 2xS-CC 1\|3 | 6.27 $^{+0.09}_{-0.14}$ | 81.6 | 29.1 | 528.8 |
| 2xS-CC 1\|6 | 6.94 $^{+0.07}_{-0.03}$ | 90.2 | 35.6 | 648.5 |
| 2xS-CC 2\|5 | 6.67 $^{+0.05}_{-0.09}$ | 86.8 | 33.8 | 615.0 |
| 2xS-CC 3\|6 | 7.02 $^{+0.16}_{-0.12}$ | 91.3 | 43.8 | 796.4 |
| 2xS-CC 4\|6 | 7.14 $^{+0.02}_{-0.02}$ | 92.9 | 47.1 | 857.3 |
| 2xS-CC 5\|7 | 7.30 $^{+0.08}_{-0.07}$ | 94.9 | 56.7 | 1031.2 |
| 2xS-CC 6\|7 | 7.40 $^{+0.13}_{-0.15}$ | 96.2 | 63.2 | 1149.5 |

**Fully predictive**   Results for speech separation using fully predictive SOI are presented in figure 5. Details for selected models can be found in table 2. Reduction of metrics observed for fully predictive variant tends to be larger than for a single S-CC layer in partially predictive SOI. Additional reduction of metrics stems from added shift in time axis. Added shift produces a pattern where some part of model can be updated between inferences, as it only depends on past data. Size of this

model part depends on the position of an SC or SS-CC layer within network. We refer to the size of this model part as "Precomputed" in table 2. The increase of precomputed partial state leads to reduction of SI-SNRi metric but allows to speed-up computation, because this part can be computed in advance. With FP SOI we achieved 50% reduction of computational cost at the expense of 11.3% of SI-SNRi but with 83.7% of network calculated using past data only.

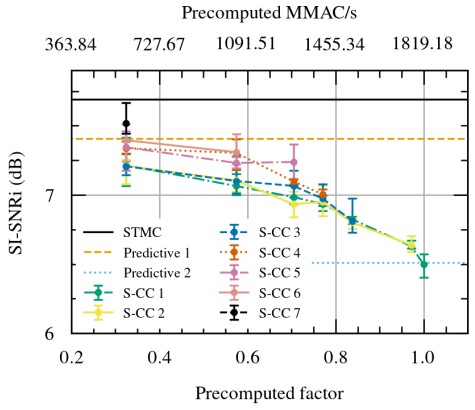

Figure 5: Results of speech separation experiment with FP SOI.

Table 2: Selected results from experiments with FP SOI for speech separation.

| Model | SI-SNRi (dB) | SI-SNRi retain (%) | Complexity retain (%) | Complexity (MMAC/s) | Precomputed (%) |
|---|---|---|---|---|---|
| STMC | **7.69** $^{+0.06}_{-0.08}$ | **100.0** | 100.0 | 1819.2 | 0.0 |
| Predictive 1 | 7.41 $^{+0.07}_{-0.09}$ | 96.3 | 100.0 | 1819.2 | **100.0** |
| Predictive 2 | 6.51 $^{+0.03}_{-0.08}$ | 84.7 | 100.0 | 1819.2 | **100.0** |
| SS-CC 2 | 6.64 $^{+0.07}_{-0.05}$ | 86.3 | 51.4 | 935.2 | 97.2 |
| SS-CC 5 | 7.24 $^{+0.13}_{-0.16}$ | 94.1 | 64.8 | 1178.7 | 70.4 |
| SS-CC 7 | 7.52 $^{+0.15}_{-0.07}$ | 97.8 | 83.8 | 1524.3 | 32.4 |
| SS-CC 1\|3 | 6.82 $^{+0.02}_{-0.04}$ | 88.7 | **50.0** | **909.6** | 83.7 |
| SS-CC 1\|6 | 7.06 $^{+0.09}_{-0.06}$ | 91.8 | **50.0** | **909.6** | 57.4 |
| SS-CC 2\|5 | 6.93 $^{+0.09}_{-0.09}$ | 90.1 | 51.4 | 935.2 | 70.4 |
| SS-CC 3\|6 | 7.10 $^{+0.18}_{-0.09}$ | 92.3 | 58.1 | 1057.5 | 57.4 |
| SS-CC 4\|6 | 7.30 $^{+0.10}_{-0.12}$ | 94.9 | 61.5 | 1118.4 | 57.4 |
| SS-CC 5\|6 | 7.23 $^{+0.05}_{-0.04}$ | 94.0 | 64.8 | 1178.7 | 57.4 |
| SS-CC 6\|7 | 7.39 $^{+0.12}_{-0.15}$ | 96.1 | 71.3 | 1296.9 | 32.4 |

## 4.2 ACOUSTIC SCENE CLASSIFICATION

Results for ASC task are collected in figure 6. Our results indicate that SOI does not lead to decrease in metrics for this particular model and task in comparison with STMC model. This can be explained by a much slower change of output (acoustic scene label) in comparison to previous task (speech mask). In our test we only compared SOI model that reduces computational cost of single inference by half and achieved results lead to conclusion that further reduction may still be possible.

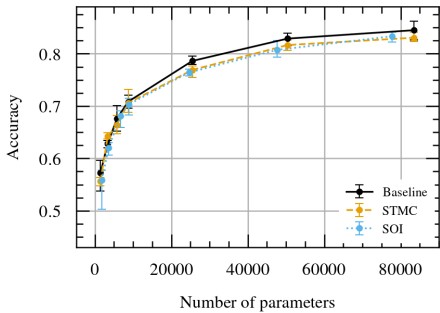

Figure 6: Results of ASC experiment.

## 5 CONCLUSION

In this work, we presented a method for reducing the computational cost of a convolutional neural network by reusing network partial states from previous inferences, leading to a generalization of these states over longer time periods. We discussed the effect of partial state prediction that our method imposes on the neural model and demonstrated that it can be applied gradually to balance model quality metrics and computational cost.

Our experiments highlight the high potential for computational cost reduction of a CNN, especially for tasks where the output remains relatively constant, such as event detection or classification. We achieved a computational cost reduction of 50% without any drop in metrics in the ASC task and a 64.4% reduction in computational cost with a relatively small reduction of 9.8% in metrics for the speech separation task. We also showcased the ability of SOI to control the trade-off between model's quality and computational cost, allowing for resource- and requirement-aware tuning.

The presented method offers an alternative to the STMC solution for strided convolution. While SOI reduces network computational complexity at the expense of measured performance, STMC ensures that metrics are not reduced but at the cost of increased memory consumption at an exponential rate. SOI is akin to methods like network pruning, but unlike pruning, it does not require special sparse kernels for inference optimization. It is worth noting that these methods are not mutually exclusive, therefore, the STMC strided convolution handler, SOI, and pruning can coexist within a neural network to achieve the desired model performance.

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

## A  TLDR: SHORT-TERM MEMORY CONVOLUTIONS (STMC)

The STMC and transposed STMC (tSTMC) layers (Stefański et al., 2023) aim to optimize convolutional neural networks by introducing a cache memory which stores past outcomes for each layer. This cache memory eliminates redundant computations by providing previously calculated values to subsequent calls of the STMC layers, which allows for efficient processing of data chunks of varying sizes. The design of STMC shift registers is based on the observation that convolutianal layers process partially overlapping input data when working in online mode. While STMC is task- and model-agnostic, it is limited to causal convolutional and pooling layers. Its effectiveness increases with the depth of the network, but it incurs an additional cost when handling strided convolutions because in such case additional states have to be stored (in additional shift registers). Figure 7 shows the diagram of STMC and tSTMC layer.

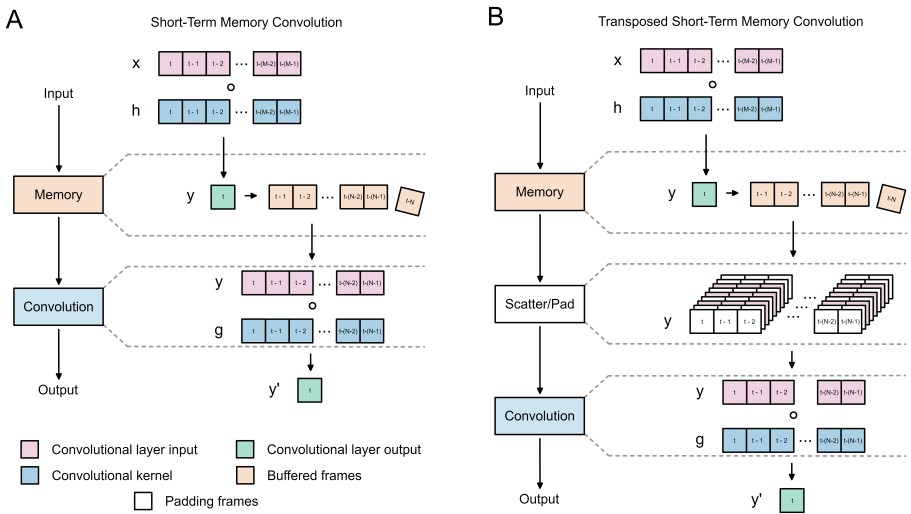

Figure 7: A) Short-Term Memory Convolution. B) Transposed Short-Term Memory Convolution.

Let's consider a 1D time series data $X_t \in \mathbb{R}^N$ comprising $N$ samples as the input of convolutional layer at time $t$. Also let us assume two convolutional kernel $h_1 \in \mathbb{R}^{M_1}$ and $h_2 \in \mathbb{R}^{M_2}$ represented by a Toeplitz matrices $H_1 \in \mathbb{R}^{N \times (N-M_1+1)}$ and $H_2 \in \mathbb{R}^{(N-M_1+1) \times (N-M_1-M_2+2)}$. Output $Z_t$ of both operations can be calculated as follows:

$$Y_t = \sigma(H_1 \cdot X_t^{\mathrm{T}}) \tag{8}$$

$$Z_t = \sigma(H_2 \cdot Y_t^{\mathrm{T}}) \tag{9}$$

where $\sigma$ is the activation function. Computational complexity of the above operations for each subsequent time step $t$ is $\mathrm{O}(NM_1) + \mathrm{O}(NM_2)$.

By adding shift register to both convolutional layers to reuse past layers outputs we can minimize computational cost as follows:

$$Y_t = (Y_{t-1} \mid \sigma(h_1 \cdot X_t^{\mathrm{T}})) \tag{10}$$

$$Z_t = (Z_{t-1} \mid \sigma(h_2 \cdot (Y_{t-1} \mid \sigma(h_1 \cdot X_t^{\mathrm{T}}))) \tag{11}$$

where $X_t \in \mathbb{R}^{M_1}$ are the newest input frames and operation done by shift register is denoted by $\mid$. The computational complexity of the STMC sequence for each time step $t$ is $\mathrm{O}(M_1) + \mathrm{O}(M_2)$ or $\mathrm{O}(KM)$.

## B    STRIDED CONVOLUTIONS ARE BETTER FOR LONGER PREDICTIONS

In this experiment, we investigated the impact of strided convolutions on predictive inference. Our test environment consisted of a U-Net model applied to a speech separation task. We examined two model variants: *Predictive* and *Strided Predictive*. The Predictive model serves as our baseline U-Net with an added time shift at the end. The Strided Predictive model, in addition to the time shift, incorporates strided convolutions in place of standard ones. For each model, we conducted tests with four different lengths of prediction, ranging from 1 frame to 4 frames. We performed five training runs for each model using the setup outlined in Section 3.1. The results of this experiment are displayed in Figure 8 and Table 3.

We conclude that strided convolutions shows higher potential for longer predictions. We attribute this effect to the fact, that using strides forces stronger generalization of outputs of strided convolutions because they are applied in multiple contexts.

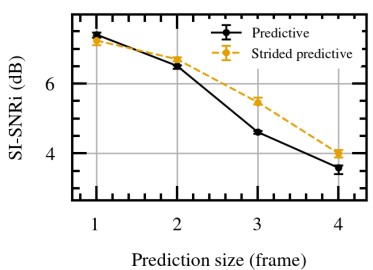

Figure 8: Comparison between standard convolutions and strided convolution in predictive inference.

Table 3: Results of experiment on the influence of strided convolution on predictive inference.

| Length of | SI-SNRi (dB) | |
|:---:|:---:|:---:|
| prediction | Predictive | Strided predictive |
| 1 | $\mathbf{7.41}^{+0.07}_{-0.09}$ | $7.24^{+0.14}_{-0.13}$ |
| 2 | $6.51^{+0.03}_{-0.08}$ | $\mathbf{6.70}^{+0.07}_{-0.05}$ |
| 3 | $4.61^{+0.04}_{-0.05}$ | $\mathbf{5.47}^{+0.14}_{-0.07}$ |
| 4 | $3.59^{+0.08}_{-0.17}$ | $\mathbf{4.00}^{+0.11}_{-0.11}$ |

## C    DIFFERENT EXTRAPOLATION METHODS

In the main paper we used element duplication as an extrapolation method but we also pointed out that any type of extrapolation may be used. Here we compare the results of our speech separation experiment using element duplication with another commonly found method – a transposed convolution. We tested both partially predictive and fully predictive modes using the setup described in Section 3.1.

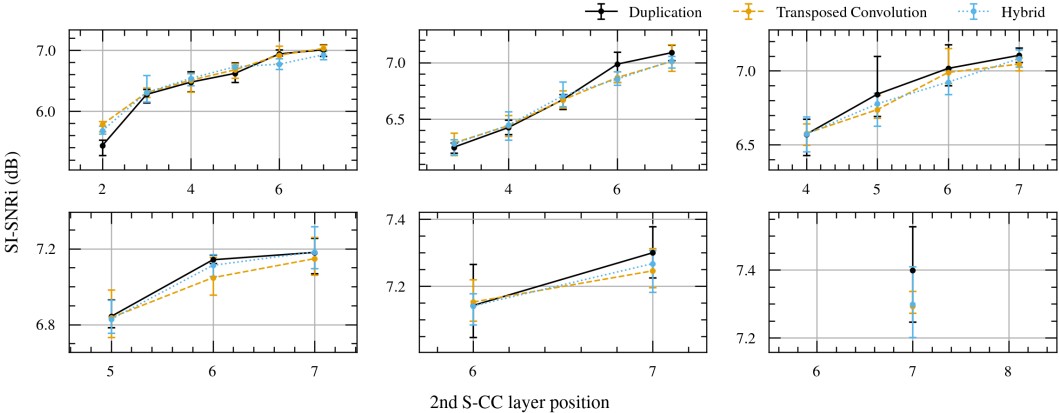

Figure 9: Comparison between frame duplication and transposed convolution for PP SOI.

Results of the experiment with PP SOI are presented in figure 9. In this case we tested a U-Net with two S-CC layers. Each plot represents a different position of the first S-CC pair and each point on X-axis represents a different position of the second S-CC pair. We also present results for hybrid models where duplication and transposed convolutions were used together – in the first and second S-CC pairs respectively.

Table 4: Results of experiment on different extrapolation methods with PP SOI.

| Model | SI-SNRi (dB) | | |
| --- | --- | --- | --- |
| | Duplication | Transposed convolution | Hybrid |
| S-CC 1\|2 | $5.43^{+0.07}_{-0.17}$ | $\mathbf{5.78}^{+0.05}_{-0.12}$ | $5.68^{+0.08}_{-0.06}$ |
| S-CC 1\|3 | $6.27^{+0.18}_{-0.14}$ | $6.30^{+0.09}_{-0.08}$ | $\mathbf{6.31}^{+0.29}_{-0.16}$ |
| S-CC 1\|4 | $6.48^{+0.18}_{-0.16}$ | $6.51^{+0.13}_{-0.20}$ | $\mathbf{6.54}^{+0.08}_{-0.11}$ |
| S-CC 1\|5 | $6.62^{+0.09}_{-0.15}$ | $6.68^{+0.11}_{-0.14}$ | $\mathbf{6.73}^{+0.03}_{-0.03}$ |
| S-CC 1\|6 | $\mathbf{6.94}^{+0.10}_{-0.04}$ | $6.92^{+0.15}_{-0.14}$ | $6.77^{+0.10}_{-0.09}$ |
| S-CC 1\|7 | $7.01^{+0.09}_{-0.11}$ | $\mathbf{7.03}^{+0.07}_{-0.03}$ | $6.92^{+0.07}_{-0.08}$ |
| S-CC 2\|3 | $6.25^{+0.04}_{-0.06}$ | $\mathbf{6.29}^{+0.09}_{-0.11}$ | $6.28^{+0.04}_{-0.10}$ |
| S-CC 2\|4 | $6.43^{+0.07}_{-0.07}$ | $6.44^{+0.10}_{-0.10}$ | $\mathbf{6.45}^{+0.12}_{-0.13}$ |
| S-CC 2\|5 | $6.67^{+0.05}_{-0.09}$ | $6.67^{+0.09}_{-0.07}$ | $\mathbf{6.71}^{+0.13}_{-0.10}$ |
| S-CC 2\|6 | $\mathbf{6.99}^{+0.11}_{-0.16}$ | $6.87^{+0.06}_{-0.05}$ | $6.85^{+0.07}_{-0.06}$ |
| S-CC 2\|7 | $\mathbf{7.09}^{+0.07}_{-0.07}$ | $7.01^{+0.15}_{-0.09}$ | $7.02^{+0.05}_{-0.07}$ |
| S-CC 3\|4 | $6.57^{+0.11}_{-0.15}$ | $\mathbf{6.58}^{+0.07}_{-0.08}$ | $6.57^{+0.12}_{-0.12}$ |
| S-CC 3\|5 | $\mathbf{6.84}^{+0.26}_{-0.15}$ | $6.74^{+0.10}_{-0.06}$ | $6.78^{+0.08}_{-0.15}$ |
| S-CC 3\|6 | $\mathbf{7.02}^{+0.17}_{-0.12}$ | $6.99^{+0.17}_{-0.08}$ | $6.92^{+0.09}_{-0.09}$ |
| S-CC 3\|7 | $\mathbf{7.10}^{+0.05}_{-0.05}$ | $7.05^{+0.05}_{-0.05}$ | $7.08^{+0.07}_{-0.06}$ |
| S-CC 4\|5 | $\mathbf{6.84}^{+0.09}_{-0.06}$ | $6.84^{+0.15}_{-0.11}$ | $6.83^{+0.11}_{-0.08}$ |
| S-CC 4\|6 | $\mathbf{7.14}^{+0.03}_{-0.18}$ | $7.05^{+0.11}_{-0.10}$ | $7.11^{+0.06}_{-0.06}$ |
| S-CC 4\|7 | $\mathbf{7.18}^{+0.08}_{-0.12}$ | $7.15^{+0.12}_{-0.09}$ | $\mathbf{7.18}^{+0.14}_{-0.09}$ |
| S-CC 5\|6 | $7.14^{+0.13}_{-0.10}$ | $\mathbf{7.15}^{+0.07}_{-0.06}$ | $7.14^{+0.04}_{-0.06}$ |
| S-CC 5\|7 | $\mathbf{7.30}^{+0.08}_{-0.08}$ | $7.25^{+0.07}_{-0.06}$ | $7.27^{+0.04}_{-0.09}$ |
| S-CC 6\|7 | $\mathbf{7.34}^{+0.13}_{-0.16}$ | $7.29^{+0.05}_{-0.03}$ | $7.30^{+0.12}_{-0.10}$ |

In this experiment neither method demonstrated a significant advantage. It seems that duplication tends to perform better than transposed convolution if it is introduced deeper within the network and *vice versa* although difference is marginal.

Results achieved with FP SOI are shown in figure 10. Each plot represents a different position of S-CC pair and each point represents different position of SC layer. Achieved results confirm previous conclusions.

This experiment proves that element duplication is a viable method and thus should be chosen for its simplicity.

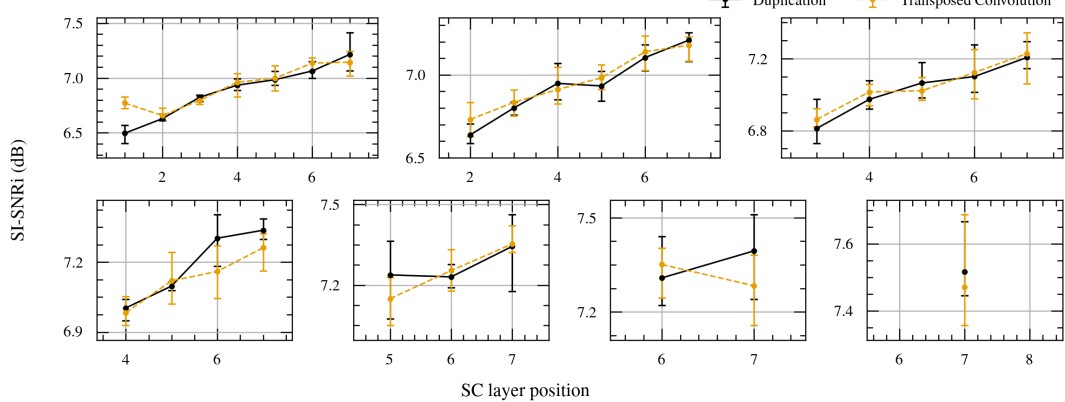

Figure 10: Comparison between frame duplication and transposed convolution for FP SOI.

Table 5: Results of experiment on different extrapolation methods with FP SOI.

| Model | SI-SNRi (dB) | |
| --- | --- | --- |
| | Duplication | Transposed convolution |
| SS-CC 1 | $6.50^{+0.08}_{-0.10}$ | $\mathbf{6.77}^{+0.06}_{-0.05}$ |
| S-CC 1\|2 | $6.63^{+0.05}_{-0.02}$ | $\mathbf{6.66}^{+0.07}_{-0.04}$ |
| S-CC 1\|3 | $\mathbf{6.82}^{+0.03}_{-0.04}$ | $6.80^{+0.04}_{-0.04}$ |
| S-CC 1\|4 | $6.93^{+0.07}_{-0.05}$ | $\mathbf{6.96}^{+0.08}_{-0.14}$ |
| S-CC 1\|5 | $6.98^{+0.08}_{-0.06}$ | $\mathbf{7.00}^{+0.12}_{-0.12}$ |
| S-CC 1\|6 | $7.06^{+0.09}_{-0.07}$ | $\mathbf{7.14}^{+0.05}_{-0.08}$ |
| S-CC 1\|7 | $\mathbf{7.21}^{+0.20}_{-0.15}$ | $7.14^{+0.11}_{-0.14}$ |
| SS-CC 2 | $6.64^{+0.05}_{-0.13}$ | $\mathbf{6.73}^{+0.11}_{-0.11}$ |
| S-CC 2\|3 | $6.80^{+0.08}_{-0.08}$ | $\mathbf{6.83}^{+0.08}_{-0.09}$ |
| S-CC 2\|4 | $\mathbf{6.95}^{+0.10}_{-0.10}$ | $6.91^{+0.14}_{-0.09}$ |
| S-CC 2\|5 | $6.93^{+0.13}_{-0.10}$ | $\mathbf{6.98}^{+0.08}_{-0.07}$ |
| S-CC 2\|6 | $7.10^{+0.04}_{-0.05}$ | $\mathbf{7.14}^{+0.10}_{-0.12}$ |
| S-CC 2\|7 | $\mathbf{7.21}^{+0.07}_{-0.06}$ | $7.18^{+0.06}_{-0.10}$ |
| SS-CC 3 | $6.81^{+0.17}_{-0.09}$ | $\mathbf{6.86}^{+0.07}_{-0.04}$ |
| S-CC 3\|4 | $6.97^{+0.11}_{-0.06}$ | $\mathbf{7.01}^{+0.05}_{-0.08}$ |
| S-CC 3\|5 | $\mathbf{7.06}^{+0.12}_{-0.09}$ | $7.02^{+0.08}_{-0.06}$ |
| S-CC 3\|6 | $7.10^{+0.18}_{-0.09}$ | $\mathbf{7.12}^{+0.13}_{-0.15}$ |
| S-CC 3\|7 | $7.21^{+0.09}_{-0.07}$ | $\mathbf{7.23}^{+0.12}_{-0.17}$ |
| SS-CC 4 | $\mathbf{7.00}^{+0.04}_{-0.06}$ | $6.98^{+0.08}_{-0.06}$ |
| S-CC 4\|5 | $7.10^{+0.02}_{-0.02}$ | $\mathbf{7.12}^{+0.13}_{-0.10}$ |
| S-CC 4\|6 | $\mathbf{7.30}^{+0.11}_{-0.13}$ | $7.16^{+0.12}_{-0.12}$ |
| S-CC 4\|7 | $\mathbf{7.34}^{+0.05}_{-0.04}$ | $7.26^{+0.07}_{-0.10}$ |
| SS-CC 5 | $\mathbf{7.24}^{+0.13}_{-0.17}$ | $7.15^{+0.08}_{-0.10}$ |
| S-CC 5\|6 | $7.23^{+0.05}_{-0.04}$ | $\mathbf{7.25}^{+0.08}_{-0.08}$ |
| S-CC 5\|7 | $7.34^{+0.12}_{-0.17}$ | $\mathbf{7.35}^{+0.07}_{-0.04}$ |
| SS-CC 6 | $7.31^{+0.14}_{-0.09}$ | $\mathbf{7.35}^{+0.06}_{-0.11}$ |
| S-CC 6\|7 | $\mathbf{7.39}^{+0.12}_{-0.16}$ | $7.28^{+0.10}_{-0.13}$ |
| SS-CC 7 | $\mathbf{7.52}^{+0.16}_{-0.08}$ | $7.47^{+0.22}_{-0.12}$ |

## D  INTERPOLATION

Our method may also benefit from the usage of interpolation methods in place of extrapolation. We tested a singular S-CC layer with three different types of interpolation: nearest-neighbor, bilinear, and bicubic. The results of this experiment can be observed in Figure 11 and Table 6. For comparison, we included our extrapolated duplication method in the results of this experiment.

In this experiment, we achieved the best results with bilinear or bicubic interpolation, although the results for bilinear interpolation showed much higher variance than any other method. It is important to remember that even though we achieve slightly better results using interpolation, the usage of interpolation comes at the cost of higher latency as we need to wait for an additional time frame.

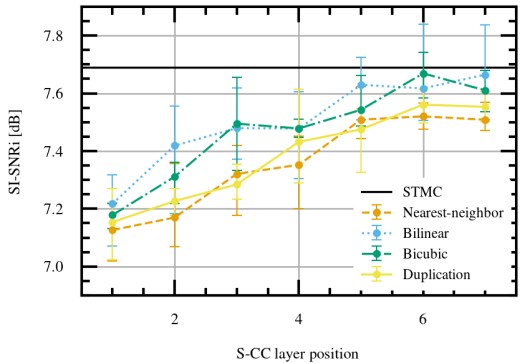

Figure 11: Comparison between extrapolated frame duplication and interpolation methods for PP SOI.

Table 6: Results of experiment on interpolation methods with PP SOI.

| Model | SI-SNRi (dB) | | | |
| --- | --- | --- | --- | --- |
| | Duplication | Nearest-neighbor | Bilinear | Bicubic |
| S-CC 1 | $7.15^{+0.12}_{-0.13}$ | $7.13^{+0.10}_{-0.11}$ | $\mathbf{7.22}^{+0.11}_{-0.15}$ | $7.18^{+0.05}_{-0.05}$ |
| S-CC 2 | $7.23^{+0.19}_{-0.11}$ | $7.17^{+0.06}_{-0.05}$ | $\mathbf{7.42}^{+0.14}_{-0.24}$ | $7.31^{+0.05}_{-0.10}$ |
| S-CC 3 | $7.28^{+0.07}_{-0.05}$ | $7.32^{+0.10}_{-0.15}$ | $7.48^{+0.14}_{-0.11}$ | $\mathbf{7.49}^{+0.17}_{-0.17}$ |
| S-CC 4 | $7.43^{+0.18}_{-0.14}$ | $7.35^{+0.11}_{-0.16}$ | $\mathbf{7.48}^{+0.13}_{-0.18}$ | $\mathbf{7.48}^{+0.04}_{-0.04}$ |
| S-CC 5 | $7.47^{+0.07}_{-0.15}$ | $7.51^{+0.12}_{-0.07}$ | $\mathbf{7.63}^{+0.10}_{-0.13}$ | $7.54^{+0.12}_{-0.06}$ |
| S-CC 6 | $7.56^{+0.05}_{-0.06}$ | $7.52^{+0.05}_{-0.05}$ | $7.62^{+0.23}_{-0.12}$ | $\mathbf{7.67}^{+0.08}_{-0.09}$ |
| S-CC 7 | $7.55^{+0.05}_{-0.05}$ | $7.51^{+0.07}_{-0.04}$ | $\mathbf{7.66}^{+0.18}_{-0.16}$ | $7.61^{+0.07}_{-0.08}$ |

## E  VIDEO ACTION RECOGNITION

Other domains can benefit from SOI, as it can be applied to any time-series data. To illustrate this, we conducted experiments using SOI for an action recognition task with video data. We utilized the HMDB-51 dataset Kuehne et al. (2011), which contains 24 fps video data of human actions split into 51 classes. We trained a popular ResNet-10 architecture Gong et al. (2022) in three variants: regular, small (with halved number of channels) and tiny (with number of channels reduced fourfold), where we replaced 3D convolutional layers with 3D STMC layers. Here, we applied SOI by introducing a skip connection between the output of block 2 and the input of block 4, optimizing block 3.

To demonstrate that SOI can work not only with STMC, we also applied it to MoViNets, which use a method called "Stream buffers". We trained two variants of MoViNets, A0 and A1, in their streaming form. SOI was applied by optimizing blocks 4 and 5. Please note that SOI can be used not only with 3D convolutions but also with their popular 2D+1 variant.

All models were trained for 100 epochs with Adam optimizer and learning rate 5e-5. Each model was trained on Nvidia A100 GPU with batch size of 16. The mean training time amounted to about about 37h.

Table 7: Results of video action recognition experiment.

| Model | Regular | | SOI | |
|---|---|---|---|---|
| | Top-1 Acc (%) | Complexity (GMAC/s) | Top-1 Acc (%) | Complexity (GMAC/s) |
| ResNet-10 | 32.63 | 48.54 | 33.34 | 40.69 |
| ResNet-10 small | 31.24 | 15.05 | 31.41 | 13.09 |
| ResNet-10 tiny | 30.46 | 5.23 | 30.90 | **4.73** |
| MoViNet A0 | 34.40 | 33.15 | 31.88 | 24.26 |
| MoViNet A1 | **35.96** | 69.77 | 32.73 | 53.92 |

Achieved results suggest that SOI is suitable for video domain as well. ResNet-10 architecture proved to be highly compatible with SOI for this task as results achieved with SOI variant of this model outperformed the regular ones. We believe that this improvement was imposed by the increase in receptive field as SOI adds additional strided convolution. The achieved reduction of complexity for this family of models was between 10-17% depending on model size. For MoViNets, the decrease of around 3% in accuracy can be spotted, although the imposed complexity reduction was higher compared to ResNet and amounted to 23-30%.

## F RESAMPLING

Simple resampling of audio signal may be used to reduce the number of calculations done by neural networks but will yield significant increase in model latency. Nonetheless in this experiment we compare SOI to four different resampling methods: SoX which is based on method proposed by Soras (2004), using Kaiser window, polyphase and linear. With this method we resampled our speech separation dataset for 16k to 8k at the input of a model and then from 8k to 16k at the output of a model. For every resampling we used our baseline model and compared it to three selected SOI models. The results are presented in table 8.

Table 8: Comparison between resampling and SOI.

| Method | SI-SNRi (dB) | Complexity (MMAC/s) |
|---|---|---|
| Linear | 3.49 | 909.6 |
| Polyphase | 5.69 | 909.6 |
| Kaiser | 5.83 | 909.6 |
| SoX | 5.77 | 909.6 |
| S-CC 5 | **7.47** | 1178.7 |
| S-CC 2 | 7.23 | 935.2 |
| S-CC 1|3 | 6.27 | **528.8** |

Our results achieved with resampling suggest that this method is not suitable for the speech separation task at least for samples that are sampled in 16k or less. SOI outperforms every tested resampling method.

# G  PRUNING

In this experiment we used unstructured global magnitude pruning (similar to Han et al. (2015)) to show how our method can be combined with pruning leading to better results than pruning on a standard model. For this experiment we chose to prune "SOI 1" and "SOI 2|6" variants of our baseline model. Each step we pruned 4096 weights from model and we report how models performed on each step. The results of this experiment are presented in Fig. 12.

These results, although preliminary, indicate that using SOI together with pruning surpasses application of pruning to STMC alone. The achieved gap between STMC and SOI models was at least around 300 MMAC/s for the same performance which is about 16% of original model. Interestingly the "SOI 2|6" surpassed the "SOI 1" model at around 6 dB SI-SNRi.

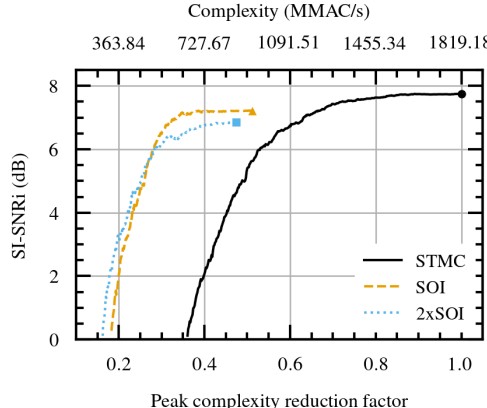

Figure 12: Pruning of STMC, SOI and 2xSOI models. Unpruned models are indicated by markers.

