# OpenReview forum: "SOI: Scaling down computational complexity by estimating partial states of the model"
_ICLR.cc/2024/Conference — Submitted to ICLR 2024_

### Official Review · Reviewer_LqQF · 2023-10-31

**Soundness:** 3 good
**Presentation:** 3 good
**Contribution:** 2 fair
**Rating:** 5
**Confidence:** 3

**Summary:**

This paper presents a novel method called Scattered Online Inference (SOI) that aims to reduce the computational complexity of Artificial Neural Networks (ANNs) while maintaining their performance. The SOI method is based on the principles of partial predictions of the network's future state, a two-phase system involving data compression using strided convolutions and reconstruction through extrapolation, achieving an RNN-like inference pattern within CNNs by leveraging the capabilities of Short-Term Memory Convolution (STMC) layers, preserving the causal nature of the optimized network architecture, and being applicable to single-frame online inference.

**Strengths:**

Originality: The proposed SOI method is a creative combination of existing ideas and techniques, such as strided convolutions, STMC layers, and partial predictions, to achieve a unique approach to reducing computational complexity in ANNs.

Quality: The authors have provided a thorough explanation of the SOI method, its underlying principles, and its implementation, as well as experimental results demonstrating the effectiveness of the method in reducing computational complexity while maintaining performance.

Clarity: The paper is well-written and organized, with clear explanations of the concepts and methods involved, making it easy for readers to understand the proposed approach and its implications.

Significance: The SOI method addresses an important challenge in the field of ANNs, particularly for real-time applications and small appliances where energy consumption and time are crucial factors. By reducing the computational complexity of ANNs, this method has the potential to enable more efficient deployment of neural networks in such scenarios.

**Weaknesses:**

The paper could benefit from a more comprehensive comparison with other existing methods for reducing computational complexity in ANNs, highlighting the specific advantages and potential limitations of the SOI method compared to alternative approaches.

While the authors have provided experimental results for the audio separation and acoustic scene classification tasks, it would be beneficial to see the performance of the SOI method in other application domains, to demonstrate its generalizability and potential impact across different types of problems.

In the experimental results presented in Table 2, the SOI method, when compared to STMC, does not show a significant difference in complexity for the closest SI-SNRi metric: STMC achieves 7.69 with 100% complexity, while Predictive 1 achieves 7.41 with 96.3% complexity. It is worth noting that a slight compromise in performance metrics can lead to a significant reduction in model complexity. Therefore, the claim of computational optimization in the abstract, which states a 64.4% reduction in computational complexity at the cost of 9.8% of SI-SNRi for the PP variant, and a 41.9% reduction at the cost of 7.70% SI-SNRi with the FP variant, may not be entirely convincing, as similar results might be achievable by simply reducing the model size without necessarily relying on the SOI method.
A more fair comparison should be made on the same performance level with different complexities or the same complexity level with different performances. From the experiments, I did not see a significant advantage of the SOI method compared to STMC in this regard.

**Questions:**

How does the SOI method perform when applied to other types of ANNs, such as Recurrent Neural Networks (RNNs) or Transformer-based models? Can the method be easily adapted to these architectures?

Are there any specific types of problems or application domains where the SOI method is particularly well-suited or might encounter difficulties? If so, could the authors provide some insights into the factors that contribute to these differences in performance?

Can the SOI method be combined with other techniques for model optimization, such as pruning or quantization, to further improve efficiency and performance? If so, are there any potential trade-offs or challenges that need to be considered when combining these methods?

---

> ### Author Response · Authors · 2023-11-23
>
> Hello! We are thankful for the constructive feedback and the pointers on how to improve on our work. We carefully studied the input of all the reviewers and made the best effort to thoroughly address the main concerns in the limited time frame. Due to the limited resources, the presented results should be treated as preliminary and we strongly believe that larger gains could be achieved after more extensive parameter tuning.
>
> The main concern regarding our research was raised about limited comparison with other methods as well as constrained application domain. We have addressed these issues by extending our experiments with video action recognition tasks utilizing ResNet and MoViNet architectures. The latter is used with so-called stream buffers, which additionally shows the possibility of applying the SOI without the STMC. The experiments showed that the application of SOI leads to the improvement of the performance for chosen task in the video domain.
>
> To improve the understanding of distinctive properties and differences of SOI and STMC we designed experiments showing that the application of strided convolutions improves the results when the predictive inference is required. It is a crucial observation because depending on the application, one may be interested in minimization of either an average or a maximal cost of the inference of a single frame, and thus inclined to choose between a partially or a fully predictive inference modes. To accustom the readers with STMC we have added an appendix with a brief overview of STMC.
>
> SOI was designed as an overlay to STMC in order to address STMC's shortcomings. We strongly believe that the model's computational complexity can be further reduced by applying the pruning method. Assuming the correctness of the Lottery Ticket Hypothesis, pruning can effectively reduce model's size without affecting the performance, and thus can be utilized as a next optimization step after application of STMC and/or SOI. Early preliminary results of such an approach are presented in appendix G. Further research on that matter is currently in progress and we intend to publish it as a separate study.
>
> Please let us answer to all your questions and concerns:
>
> > The paper could benefit from a more comprehensive comparison with other existing methods for reducing computational complexity in ANNs, highlighting the specific advantages and potential limitations of the SOI method compared to alternative approaches.
>
> > In the experimental results presented in Table 2, the SOI method, when compared to STMC, does not show a significant difference in complexity for the closest SI-SNRi metric: STMC achieves 7.69 with 100% complexity, while Predictive 1 achieves 7.41 with 96.3% complexity. It is worth noting that a slight compromise in performance metrics can lead to a significant reduction in model complexity. Therefore, the claim of computational optimization in the abstract, which states a 64.4% reduction in computational complexity at the cost of 9.8% of SI-SNRi for the PP variant, and a 41.9% reduction at the cost of 7.70% SI-SNRi with the FP variant, may not be entirely convincing, as similar results might be achievable by simply reducing the model size without necessarily relying on the SOI method. A more fair comparison should be made on the same performance level with different complexities or the same complexity level with different performances. From the experiments, I did not see a significant advantage of the SOI method compared to STMC in this regard.
>
> We hope that the added experiments in Appendices E-G will strengthen our paper. Please note that there are not a lot of methods that we can compare against as SOI can work alongside most (if not all) of state-of-the-art methods, like any type of pruning.
>
> > While the authors have provided experimental results for the audio separation and acoustic scene classification tasks, it would be beneficial to see the performance of the SOI method in other application domains, to demonstrate its generalizability and potential impact across different types of problems.
>
> We added the experiment with an application of SOI to the video action recognition task in Appendix E.
>
> > How does the SOI method perform when applied to other types of ANNs, such as Recurrent Neural Networks (RNNs) or Transformer-based models? Can the method be easily adapted to these architectures?
>
> Please be informed that our method can work in RNN. In fact we have the RNN-based model that works with SOI but we are planning to release it separately as in that model there is much more going on than just SOI. Another reason to not include RNN in our current research is that this paper already has 19 pages and we would like keep it concise and easy to understand.

---

> > ### Author Response · Authors · 2023-11-23
> >
> > > Are there any specific types of problems or application domains where the SOI method is particularly well-suited or might encounter difficulties? If so, could the authors provide some insights into the factors that contribute to these differences in performance?
> >
> > We work in audio domain and we can confirm that for this specific domain SOI works well. Our results in video domain suggest that it might be a great tool for it as well but we don’t have much experience with video analysis. We gladly invite other researchers to build their solutions upon SOI to further test its applicability to other domains.
> >
> > > Can the SOI method be combined with other techniques for model optimization, such as pruning or quantization, to further improve efficiency and performance? If so, are there any potential trade-offs or challenges that need to be considered when combining these methods?
> >
> > Yes, in fact with most of them (if not all). We added pruning experiment in Appendix G.

---

### Official Review · Reviewer_iRws · 2023-11-01

**Soundness:** 3 good
**Presentation:** 3 good
**Contribution:** 2 fair
**Rating:** 5
**Confidence:** 3

**Summary:**

The paper addresses the challenge of deploying artificial neural networks (ANNs) on small consumer electronics, such as wireless in-ear headphones, smartwatches, and AR glasses. Despite advancements in Microcontroller Units (MCUs), these devices still struggle to run state-of-the-art ANNs efficiently, especially in time-sensitive scenarios.
In this work, the main contribution goes to
1. Introduction of Scattered Online Inference (SOI):
- SOI significantly reduces the computational complexity of artificial neural networks, especially in deeper layers, by leveraging the continuity and seasonality in time-series data for extrapolation and processing speed improvements.
- The method enables real-time processing of time-series data element by element, transforming conventional CNN models for online inference and addressing challenges associated with strided convolutions.
2. Efficiency and Performance in Real-Time Systems:
- SOI offers substantial computational savings with two patterns of inference: a 64.4% reduction in computational complexity for Partially Predictive (PP) and a 41.9% reduction (plus an additional 28.7% reduction in inference time) for Fully Predictive (FP), with minimal impact on performance.
- The method is particularly beneficial for small consumer electronics in real-time scenarios, providing a solution for applying state-of-the-art DNNs without extensive model compression.
3. Minimal Architectural Changes with Ecological and Economic Impact:
- SOI achieves its efficiency gains with minimal alterations to the network architecture, making it versatile for various applications where time and energy consumption are critical.
- The method addresses the need for energy-efficient neural systems, highlighting its importance from both ecological and economic perspectives.

**Strengths:**

**Originality:**
The paper introduces Scattered Online Inference (SOI), a novel method that stands out for its innovative approach to reducing computational complexity in artificial neural networks, particularly for real-time applications on small devices. Building upon the Short-Term Memory Convolution (STMC) technique, SOI provides a unique treatment of strided layers and introduces a new inference pattern, showcasing a commendable level of originality. The method’s ability to transform conventional CNN models for online inference while addressing challenges associated with strided convolutions further emphasizes its inventive nature.

**Quality:**
The quality of the paper is evident in its comprehensive approach to addressing a significant problem in the field of artificial neural networks. The authors present two distinct patterns of inference, Partially Predictive (PP) and Fully Predictive (FP), and provide quantitative results demonstrating substantial reductions in computational complexity alongside minimal impact on performance. The method’s applicability to real-time systems and small consumer electronics, as well as its consideration of energy efficiency, reflect a high standard of research quality.

**Clarity:**
The paper is well-structured, providing a clear introduction to the problem, a detailed description of the SOI method, and a presentation of the results achieved. The authors succeed in explaining complex concepts in an accessible manner, making the paper comprehensible to a broad audience.

**Significance:**
SOI’s potential impact is substantial, particularly in the realm of small consumer electronics and real-time systems where computational efficiency is paramount. The method’s ability to achieve significant reductions in computational complexity without extensive model compression is a notable advancement, addressing a critical need in the field. Furthermore, the paper’s emphasis on energy efficiency and its ecological and economic implications highlight the broader significance of the research, contributing to the ongoing discourse on sustainable and efficient computing practices.

**Weaknesses:**

**Limited Comparison:** The paper could be strengthened by including a more comprehensive comparison with other state-of-the-art methods would further enhance the paper's quality.

**Generalization:**

- *Focus on CNNs:* The paper primarily discusses the application of SOI to convolutional neural networks (CNNs). While this is a significant contribution, the generality of the method could be questioned if it is not easily adaptable or applicable to other neural network architectures. And for CNN, the only baseline lies on STMC which seems not generalizable.

- *Limited to Audio-Related Tasks*: The paper provides examples and results primarily related to audio separation and acoustic scene classification tasks. Demonstrating the effectiveness of SOI across a wider variety of use cases and domains would strengthen its generality and appeal to a broader audience.

**Questions:**

9.8% reduction in metrics of speech spearation task seems significant to me, could you provide some context to justify this reduction is minor? for example, you can claim in another paper, their performance drop is much more like xx.x% when complexity is 50% so that reader can know 9.8% is minor in a rought impression.

---

> ### Author Response · Authors · 2023-11-23
>
> Hello! We are thankful for the constructive feedback and the pointers on how to improve on our work. We carefully studied the input of all the reviewers and made the best effort to thoroughly address the main concerns in the limited time frame. Due to the limited resources, the presented results should be treated as preliminary and we strongly believe that larger gains could be achieved after more extensive parameter tuning.
>
> The main concern regarding our research was raised about limited comparison with other methods as well as constrained application domain. We have addressed these issues by extending our experiments with video action recognition tasks utilizing ResNet and MoViNet architectures. The latter is used with so-called stream buffers, which additionally shows the possibility of applying the SOI without the STMC. The experiments showed that the application of SOI leads to the improvement of the performance for chosen task in the video domain.
>
> To improve the understanding of distinctive properties and differences of SOI and STMC we designed experiments showing that the application of strided convolutions improves the results when the predictive inference is required. It is a crucial observation because depending on the application, one may be interested in minimization of either an average or a maximal cost of the inference of a single frame, and thus inclined to choose between a partially or a fully predictive inference modes. To accustom the readers with STMC we have added an appendix with a brief overview of STMC.
>
> SOI was designed as an overlay to STMC in order to address STMC's shortcomings. We strongly believe that the model's computational complexity can be further reduced by applying the pruning method. Assuming the correctness of the Lottery Ticket Hypothesis, pruning can effectively reduce model's size without affecting the performance, and thus can be utilized as a next optimization step after application of STMC and/or SOI. Early preliminary results of such an approach are presented in appendix G. Further research on that matter is currently in progress and we intend to publish it as a separate study.
>
> Please let us answer to all your questions and concerns:
>
> > Limited Comparison: The paper could be strengthened by including a more comprehensive comparison with other state-of-the-art methods would further enhance the paper's quality.
>
> We believe that added experiments in Appendices E-G will strengthen our paper. Please note that there are not a lot of methods that we can compare to as the SOI will work alongside most (if not all) of state-of-the-art methods like any type of pruning.
>
> > Focus on CNNs: The paper primarily discusses the application of SOI to convolutional neural networks (CNNs). While this is a significant contribution, the generality of the method could be questioned if it is not easily adaptable or applicable to other neural network architectures. And for CNN, the only baseline lies on STMC which seems not generalizable.
>
> We believe that the contribution to just CNNs is still substantial. Please be informed that our method can work in RNN, in fact we have the RNN-based model that work with SOI but we are planning to release it separately as in this model there is much more going on than just SOI. Another reason to not include RNN is that this paper already has 19 pages and we would like keep it concise and easy to understand.
>
> > Limited to Audio-Related Tasks: The paper provides examples and results primarily related to audio separation and acoustic scene classification tasks. Demonstrating the effectiveness of SOI across a wider variety of use cases and domains would strengthen its generality and appeal to a broader audience.
>
> We added video action recognition task in Appendix E.
>
> > 9.8% reduction in metrics of speech spearation task seems significant to me, could you provide some context to justify this reduction is minor? for example, you can claim in another paper, their performance drop is much more like xx.x% when complexity is 50% so that reader can know 9.8% is minor in a rought impression.
>
> SOI is quite adaptable and can provide the range of complexity reductions at different cost. In our case those 9.8% of SI-SNRi do not contribute significantly to the output of a model and from our experience and experiments such difference is not detectible by a human listeners during blind tests (in this specific case).

---

### Official Review · Reviewer_o3ho · 2023-11-05

**Soundness:** 3 good
**Presentation:** 3 good
**Contribution:** 3 good
**Rating:** 6
**Confidence:** 3

**Summary:**

The authors proposed a method for reducing the computational cost of a convolutional neural
network by reusing network partial states from previous inferences, leading to a generalization of
these states over longer time periods. The authors claimed that the proposed method achieves a computational cost reduction of 50% without any drop in metrics in the ASC task and a 64.4% reduction in computational cost with only a 9.8% reduction in metrics for the speech separation task. After the comparison, the paper implied that the proposed method offers an alternative to the STMC solution for strided convolution.

**Strengths:**

The authors introduced two possible patterns of inference achievable
with SOI - Partially Predictive (PP) and Fully Predictive (FP). For the audio separation task,  64.4% reduction was achieved in computational complexity at the cost of 9.8% of SI-SNRi for the PP variant, and a 41.9% reduction at the cost of 7.70% SI-SNRi with the FP variant. Moreover, the latter variant reduces inference time by an additional 28.7%. Similar results are also presented for the acoustic
scene classification task with a model based on the GhostNet architecture. From this sense, I personally believe that the authors' work provides another option for reducing time cost for time-series data processing with reasonable explanation through the details of the experiment results and corresponding theoretical analytics.

**Weaknesses:**

I would suggest adding at least two more experiments to the experiment part to strengthen and demonstrate the advantages of the proposed method. Although the current experiment is cool, to better support the authors' theory, it is always good to shoot some or more real application cases with sound results.

**Questions:**

no more question

**Details Of Ethics Concerns:**

no concerns

---

> ### Author Response · Authors · 2023-11-23
>
> Hello! We are thankful for the constructive feedback and the pointers on how to improve on our work. We carefully studied the input of all the reviewers and made the best effort to thoroughly address the main concerns in the limited time frame. Due to the limited resources, the presented results should be treated as preliminary and we strongly believe that larger gains could be achieved after more extensive parameter tuning.
>
> The main concern regarding our research was raised about limited comparison with other methods as well as constrained application domain. We have addressed these issues by extending our experiments with video action recognition tasks utilizing ResNet and MoViNet architectures. The latter is used with so-called stream buffers, which additionally shows the possibility of applying the SOI without the STMC. The experiments showed that the application of SOI leads to the improvement of the performance for chosen task in the video domain.
>
> To improve the understanding of distinctive properties and differences of SOI and STMC we designed experiments showing that the application of strided convolutions improves the results when the predictive inference is required. It is a crucial observation because depending on the application, one may be interested in minimization of either an average or a maximal cost of the inference of a single frame, and thus inclined to choose between a partially or a fully predictive inference modes. To accustom the readers with STMC we have added an appendix with a brief overview of STMC.
>
> SOI was designed as an overlay to STMC in order to address STMC's shortcomings. We strongly believe that the model's computational complexity can be further reduced by applying the pruning method. Assuming the correctness of the Lottery Ticket Hypothesis, pruning can effectively reduce model's size without affecting the performance, and thus can be utilized as a next optimization step after application of STMC and/or SOI. Early preliminary results of such an approach are presented in appendix G. Further research on that matter is currently in progress and we intend to publish it as a separate study.
>
> Please let us answer to all your questions and concerns:
>
> > I would suggest adding at least two more experiments to the experiment part to strengthen and demonstrate the advantages of the proposed method. Although the current experiment is cool, to better support the authors' theory, it is always good to shoot some or more real application cases with sound results.
>
> We added 3 new sections to appendix.
>
> In Appendix E, we added experiment with action recognition task from video data to show that our method works well in other domains. In this section we also applied SOI to MoViNets which we used with their original ‘stream buffers’ method instead of STMC to show that it can also work with methods similar to STMC.
>
> In Appendix F, we compare our method to resampling using common resampling methods.
>
> In Appendix G, we show that our method works well with pruning which is commonly used in our scenario.

---

### Official Review · Reviewer_sLXH · 2023-11-09

**Soundness:** 2 fair
**Presentation:** 2 fair
**Contribution:** 2 fair
**Rating:** 5
**Confidence:** 3

**Summary:**

The authors present a method called scattered online inference that applies strides to convolutional processing of time series data, and performing partial or full prediction for extrapolating values in between. The authors test this method on a number of benchmarks and show that their method can get close to the performance of STMC.

**Strengths:**

The method seems to provide significant computational savings without too big a big drop in performance.

**Weaknesses:**

I found it very hard to piece together what the authors have done in this paper: the problem being solved, the specific application domain, the specific format of the input, the differences from STMC etc. The authors seem to depend entirely on the STMC and other papers to provide this context. I had to read a number of previous papers to understand even the context of the paper. The authors need to give more background in the main text.

The empirical comparisons also seem very limited, including only STMC and some variants of their method.

**Questions:**

There are various presentation issues. Examples:
- Fig. 1: C and E seem identical.
- Sec. 2.2: X seems to have length N and N also seems to be the number of samples.
- in Abstract, second to last sentence: Not clear what latter refers to
- Fig. 2 caption has two B). Seems to be off by one.

---

> ### Author Response · Authors · 2023-11-23
>
> Hello! We are thankful for the constructive feedback and the pointers on how to improve on our work. We carefully studied the input of all the reviewers and made the best effort to thoroughly address the main concerns in the limited time frame. Due to the limited resources, the presented results should be treated as preliminary and we strongly believe that larger gains could be achieved after more extensive parameter tuning.
>
> The main concern regarding our research was raised about limited comparison with other methods as well as constrained application domain. We have addressed these issues by extending our experiments with video action recognition tasks utilizing ResNet and MoViNet architectures. The latter is used with so-called stream buffers, which additionally shows the possibility of applying the SOI without the STMC. The experiments showed that the application of SOI leads to the improvement of the performance for chosen task in the video domain.
>
> To improve the understanding of distinctive properties and differences of SOI and STMC we designed experiments showing that the application of strided convolutions improves the results when the predictive inference is required. It is a crucial observation because depending on the application, one may be interested in minimization of either an average or a maximal cost of the inference of a single frame, and thus inclined to choose between a partially or a fully predictive inference modes. To accustom the readers with STMC we have added an appendix with a brief overview of STMC.
>
> SOI was designed as an overlay to STMC in order to address STMC's shortcomings. We strongly believe that the model's computational complexity can be further reduced by applying the pruning method. Assuming the correctness of the Lottery Ticket Hypothesis, pruning can effectively reduce model's size without affecting the performance, and thus can be utilized as a next optimization step after application of STMC and/or SOI. Early preliminary results of such an approach are presented in appendix G. Further research on that matter is currently in progress and we intend to publish it as a separate study.
>
> Please let us answer to all your questions and concerns:
>
> > I found it very hard to piece together what the authors have done in this paper:
>
> > -	the problem being solved,
>
> Reduction of computational complexity of NN models for real-time processing without imposing additional latency.
>
> > -	the specific application domain,
>
> Time-series data, Real-time processing. Beside that there is no specific domain.
>
> > -	the specific format of the input,
>
> The specific format of the input depends on the used network but in all cases it needs to be time-series. In our speech separation task and ASC task we used spectrograms in BCFT format (B-batch, C-channels, F-frequency, T-time).
>
> > The authors seem to depend entirely on the STMC and other papers to provide this context. I had to read a number of previous papers to understand even the context of the paper. The authors need to give more background in the main text.
>
> Adding more context in main text is impossible as there is the 9 page limit for main text. Therefore we added more context on STMC in Appedix A and referenced it in Introduction so readers can familiarize themselves with STMC to the level that should be sufficient to understand the paper. In Appendix E, we also showed that this method does not entirely depend on STMC as we applied it with ‘stream buffers’ method in place of STMC.
>
> > The empirical comparisons also seem very limited, including only STMC and some variants of their method.
>
> In appendix E, we added experiment with action recognition task from video data using ResNet-10 architecture and MoViNets in A0 and A1 variants. This experiment also proves that our method is not dependent on STMC as with MoViNets utlize ‘stream buffers’.
>
> In appendix F we added comparison of our method to common resampling methods which can be used to reduce computational complexity as well.
>
> In appendix G we show how our method works with pruning.
>
> > Fig. 1: C and E seem identical.
>
> Fig. 1C and Fig. 1E are not identical as operation in Fig. 1E has an additional shift in time represented by relative position of time frames in time axis (time axis is also depicted). Position of frames in time is important as it defines how we interpret them throughout the network and at output. As this was not clear we added cropping to Fig. 1E to make it easier to follow.
>
> > Sec. 2.2: X seems to have length N and N also seems to be the number of samples.
>
> Yes, 1D vector X Is composed of N samples and that’s why it has length N.
>
>
> > in Abstract, second to last sentence: Not clear what latter refers to
>
> Modified to clearly state that this sentence refers to FP variant of SOI
>
> > Fig. 2 caption has two B). Seems to be off by one.
>
> Fixed. Thank you!

---

### Meta-Review · Area_Chair_Cw43 · 2023-12-08

**Metareview:**

The paper introduces Scattered Online Inference (SOI) which aims to reduce the computational cost in artificial neural networks. A main technique is to partial predict the network's future states.

The topic is interesting and important. However, the reviewers noted that the experimental evaluation is rather limited and constrained to particular tasks. Given the empirical nature, a more thorough experimental evaluation on a wide range of architectures and problems is definitely needed. Further, reviewers also remarked the the exposition of the paper could be improved.

**Justification For Why Not Higher Score:**

This is an important topic, but given the empirical nature of the approach, a much more thorough evaluation is need to demonstrate the method's efficacy.

**Justification For Why Not Lower Score:**

NA

---

### Decision · Program_Chairs · 2024-01-16

Reject